# An Experimental Study on the Healing Performance of Complex Capsules Using Multiphase Inorganic Materials for Crack Self-Healing of Cement Mortars

**DOI:** 10.3390/ma15248819

**Published:** 2022-12-09

**Authors:** Yun-Wang Choi, Cheol-Gyu Kim, Eun-Joon Nam, Sung-Rok Oh

**Affiliations:** Department of Civil Engineering, Semyung University, 65 Semyeong-ro, Jecheon-si 27136, Republic of Korea

**Keywords:** cement mortar, complex capsules, crack, healing performance, inorganic materials, self-healing, multiphase

## Abstract

Recently, a self-healing technique capable of repairing cracks in structures has emerged. Among various self-healing technologies, self-healing capsules can be largely classified into two types, depending on the phase of the core material: solid capsules, in which the core material is a powder; and microcapsules, in which the core material is a liquid. Solid capsules and microcapsules have different mechanisms, and their capsule sizes are also distinctly different. This suggests that each has advantages and disadvantages. Most of the studies known to date have utilized single capsules. However, if one uses a mixture of the two types of capsules, it is possible to highlight the strengths of each capsule and compensate for the weaknesses. Therefore, in this study, the first research on complex capsules that mixed solid capsules and microcapsules was attempted. As a result of the experiment, the complex capsule slightly reduced the fluidity of the mortar, but the effect was not significant. Moreover, the complex capsule tended to reduce the compressive strength of the mortar. In particular, it was found that the effect of solid capsules on the reduction in compressive strength among complex capsules was greater than that of microcapsules. Conversely, the healing performance increased when the ratio of solid capsules in the complex capsules was large.

## 1. Introduction

Recently, self-healing technology has been gaining popularity for reducing and repairing cracks in structures, especially in developed countries. Self-healing technology can heal cracks on their own without any additional action in the early stages of cracking. Accordingly, it is possible to greatly reduce the cost, effort, and time required for general structural maintenance. In particular, since it is effective even when it is difficult for people to access [1,2], this is considered a very effective smart technology in the structural repair industry. Crack self-healing technology has been initially studied in developed countries such as the UK, USA, and Japan [3,4], and recently research has also been steadily reported in Korea, India, and China [5,6,7,8,9].

There are various methods for applying self-healing technology to structures. In terms of materials, there are typically bacteria and capsules, while in terms of construction, there is a method of coating the surface of a structure or mixing it with concrete. There are numerous results showing that bacteria have excellent healing properties. However, from a long-term perspective, it is difficult to expect healing performance due to the limitation of food supply and demand for bacteria. A method using a capsule can compensate for these technical limitations and disadvantages.

In the capsule, the core material—which is a healing material—is protected by the capsule membrane. Since the capsule reacts only when it is destroyed by cracking, it is intact until cracking occurs. Accordingly, the healing energy of the capsule is semi-permanent. These features bring the advantage of being able to respond to long-term cracking. This self-healing technology using capsules has two applications: the capsule can be mixed with the structure’s finishing material to coat the concrete surface, or it can be mixed with the concrete mixture to be used as a concrete matrix. In the former case, the cost is low because the amount of capsules used is relatively small, but the healing performance is limited to the surface of the structure. In the latter case, the cost is relatively high due to the large amount of capsules used, but the healing performance is extended to the entire structural matrix. However, crack repair is more important than cost for structures with high importance or that are difficult to access for personnel. Therefore, the method of applying capsules for self-healing of cracks should be applied with comprehensive consideration of environmental and economic benefits. In this paper, as a cornerstone study for mixing capsules into concrete mixtures, the effects of capsules on mortar quality and healing performance were evaluated.

The capsule used in this study was a complex capsule. The complex capsule defined in this paper refers to a mixture of capsules with two phases. The self-healing technology using capsules known so far is largely divided into two types [7,8,9,10,11,12]: solid capsules using a powdery core material, and liquid capsules using a liquid core material. Liquid-type capsules are mainly called micro- or nanocapsules. The majority of the studies to date have used microcapsules [8,9,10,12], and recently studies using solid capsules [7,11] have been introduced. Solid capsules and microcapsules have different reaction mechanisms for healing and have different capsule size ranges. Accordingly, each capsule has advantages and disadvantages. The capsules used in this study were complex capsules involving both solid capsules and microcapsules. The size of the solid capsules was about 600~850 μm, and their core material reacts to moisture and heals the crack through crystal growth [9,10]. The size of the microcapsules was 50~300 μm, and their core material reacts to moisture and heals cracks via silicate-based reactants.

Looking at capsule-related studies, it has been reported that solid capsules have higher healing efficiency than microcapsules. However, since the size of solid capsules is relatively larger than that of microcapsules, the amount of capsules present on the crack surface is small. Therefore, it is difficult to expect healing performance in the part without the capsules. In addition, since the solid capsule is a mixture material that cannot receive a load, there is a concern about a decrease in strength. These problems can be partially solved by using microcapsules. Since microcapsules are relatively smaller than solid capsules, healing performance can be expected in microcracks, but it is difficult to expect healing performance in large cracks. Therefore, when solid capsules and microcapsules are used in combination, the disadvantages of each capsule can be overcome.

When solid capsules and microcapsules are mixed and used together, the effect of correcting the emptiness of capsule particles can be obtained due to the different particle sizes. In addition, calcium hydroxide generated from solid capsules can be used as a reaction catalyst for microcapsules and can promote healing reactions. Conversely, silicate-based products derived from microcapsules can promote crystal growth of the healing products in solid capsules and promote hydration of unhydrated cement. In this case, the decrease in strength can be minimized and the healing performance can be improved. In this way, the solid capsule and the microcapsule can highlight one another’s strengths and compensate for their weaknesses. However, most studies introduced to date have only used each capsule type alone. Therefore, in this study, as a basic investigation of the mixed use of solid capsules and microcapsules, the quality characteristics and healing performance of cement mortar using composite capsules were evaluated. Through the results of this paper, the optimal mixing ratio of solid capsules and microcapsules in complex capsules is to be considered.

## 2. Materials and Methods

### 2.1. Complex Capsules Using Multiphase Inorganic Materials

The composite capsules used in this study were obtained by mixing self-healing solid capsules (SCs) and self-healing microcapsules (MCs) [9,10]. The core material of SCs is a mixture of an expansion material (calcium sulfoaluminate (CSA)) and anhydrous gypsum (CaSO_4_) at a certain ratio. For the ratio of CSA and CaSO_4_, the optimal ratio obtained through a previous study [9] was applied. Figure 1a shows a sample of SCs, while Figure 1b shows the reaction mechanism of SCs [13]. The core material of the MCs is a liquid-type silicate-based material, which is a mixture of sodium silicate, potassium silicate, and lithium silicate at a certain ratio. For the ratio of silicate-based materials, the optimal ratio obtained through previous research [14] was applied. Figure 2a shows a sample of MCs, while Figure 2b shows the reaction mechanism of the MCs. Table 1 shows the characteristics and manufacturing method of each capsule, while Figure 3 shows the reaction mechanism of the complex capsule. 

### 2.2. Cement Mortar

Table 2 shows the mixing ratio of the cement mortar. Ordinary Portland cement (C) (Hanil, Jecheon-city, Republic of Korea) was used for the cement, and ISO standard sand (S) was used for the fine aggregate. Table 3 shows the particle size distribution of the ISO standard sand. In addition, in order to secure the target fluidity of the cement mortar, a polycarboxylic high-performance water-reducing agent (ad.) was used, which can reduce the loss of capsules by improving the workability of the mortar and facilitates the dispersion of the capsules. Complex capsules (CCs) were mixed at 3% based on the mass of the binder (B). The composition ratio of the CCs was ‘SCs:MCs’ at three levels (3:7, 5:5, and 7:3).

### 2.3. Test Methods

#### 2.3.1. Fresh State Test

The flow of the mortar was measured using a flow table and flow cone according to ASTM C1437 and ISO 679 [15]. Figure 4a shows the mortar flow test. The mortar flow was measured in three directions, and then the average value was obtained.

#### 2.3.2. Hardening State Test

The compressive strength of the mortar was measured according to ASTM C109 and ISO 679 [16]. The flexural strength of the mortar was measured according to ASTM C78 and ISO 679 [17]. Figure 4b shows the compressive strength test, and Figure 4c shows the flexural strength test. According to ISO 679, a test piece of 40 × 40 × 160 mm was prepared and the flexural strength was measured first. After the flexural strength was measured, the remaining two pieces were used as test pieces for measuring the compressive strength. Therefore, as for the number of test pieces, three flexural strength test pieces were molded, and compressive strength measurements were performed using 6 fractured pieces of the flexural strength test piece. A general hydraulic universal testing machine (UTM) was used as the equipment for strength testing.

#### 2.3.3. Healing Performance

The healing performance was evaluated through the water permeability test. A constant water head permeability test was adopted to measure the water flow rate of the crack-induced specimen and evaluate the self-healing performance of the repair mortars [18]. For the water permeability test, cracked specimens were produced through several steps. First, mortar cylinders (Ø100 × 200 mm) were prepared, as shown in Figure 5a. Next, these cylinders were demolded after 24 h. They were cured in a water bath at 20 °C until they reached the crack induction stage. 

Once the cracking stage was attained, a cylinder was sliced into three disc specimens (Ø100 × 50 mm). The cylinder was then split into two semicircular sections, as shown in Figure 5b. Then, a flexible silicone rubber sheet with varying thickness was attached to both ends of the cracked sections to induce a crack of a specific width, as shown in Figure 5c. The actual lengths of the cracks were approximately 70 mm. Finally, the split specimens were bound together using stainless steel bands to maintain the desired crack widths, as shown in Figure 5d. For each specimen, cracks were induced at 28 and 91 days, with crack widths ranging from 0.2 to 0.25 mm. After the specimens were prepared, the widths and lengths of the cracks were measured using a microscope (EGVM-35M, EG Tech, Jecheon-city, Republic of Korea).

The cracked specimens were cured in a water bath at 20 °C during the healing period. As there is no set standard for the water temperature during the healing period, a water curing temperature of 20 °C was adopted [18,19]. The water permeability test was conducted after healing periods of 0, 7, 14, 21, and 28 days. Figure 6a shows the test schematic diagram of the water permeability test. Figure 6b shows the water permeability test apparatus for the cracked disc specimens. The amount of water coming out of the test equipment was measured for 7 min after stabilizing the water head and water flow. The water flow rate in units of mL/(mm·min) was determined by dividing the amount of discharged water by the test duration (min) and crack length (mm).

Through the water permeability test, the healing index (SHq) can be calculated using the water flow rate reduction ratio, as follows [18,20,21,22,23]:(1)$$SH_{q}=\left[1-\frac{q\left(t\right)}{q0}\right]\times 100\;\left(\%\right),$$
where *q*0 is the initial water flow rate measured just after the specimen is cracked, without any healing effect, while *q*(*t*) is the water flow rate after a healing period *t*.

#### 2.3.4. Crack Monitoring

Crack closure was observed at a magnification of ×100 using the same test piece as for the water flow specimen. Healing products in cracks were also observed. Figure 7 shows the front view of the crack closure test.

## 3. Results and Discussion

### 3.1. Fundamental Properties

#### 3.1.1. Fresh State Properties

Table 4 summarizes the fresh state properties of the cement mortars. The plain mortar’s flow was about 210 mm. The flows of CCs-1, CCs-2, and CCs-3 were approximately 200 mm, 195 mm, and 190 mm respectively. These results showed that the flow tended to decrease as the proportion of SCs among the CCs increased. According to previous studies, SCs report a loss of about 5% [9]. Therefore, it was observed that the flow reduction occurred due to the destruction of the SCs. CCs-3, with the largest amount of SCs, decreased by about 9.5% compared to the plain mortar’s flow. The flow loss was about 10% after 30 min, regardless of the mix type, and about 20% after 60 min, showing no significant results.

#### 3.1.2. Properties of the Hardening State

Table 5 summarizes the hardened state properties of the cement mortars. The compressive strength of the plain mortar at 28 days of age was about 50 MPa, while the compressive strength of CCs-1, CCs-2, and CCs-3 at 28 days of age was about 48 MPa, 45 MPa, and 39 MPa, respectively. All CCs showed a tendency to decrease compared to the plain mortar, with their compressive strength decreasing proportionately as the proportion of SCs among CCs increased. The compressive strength of CCs-3, with the largest proportion of SCs, decreased by about 22%. According to a previous study, SCs are the cause of the decrease in compressive strength, because they are larger than the particles of MCs and are equivalent to simple fillers that cannot receive loads [13,14]. In addition, MCs are reported to have little effect on compressive strength. Therefore, analysis indicates that the compressive strength reduction rate increased as the specific gravity of the SCs increased. In the case of the flexural strength, the same trend as the compressive strength result was shown, and it was concluded that the strength decreased due to the increase in the SCs’ interface as the number of SCs with larger particles increased.

### 3.2. Healing Performance

#### 3.2.1. Water Flow Rate

Figure 8a shows the healing rate of the 28-day healing period of the specimen with cracks induced at 7 days of age, while Figure 8b shows the healing rate of the 28-day healing period of the specimens with cracks induced at 28 days of age. As mentioned in Section 2.3.3, the results shown in Figure 8 and Figure 9 measure the initial water flow rate immediately after crack induction, followed by the changes in the water flow rate after the healing period measured at 7-day intervals. The healing rate is derived from the reduction in the water flow rate by substituting the measured result into Equation (1). According to previous studies [9], the initial water flow rate can be converted into an equivalent crack width [5,23]. Although there are some differences according to the conditions of the specimen, the crack width is 0.3 mm when the initial water flow rate is 1.0~1.8 mL/min·mm.

As shown in Figure 8a, the healing rates corresponding to a 0.3 mm crack width were approximately 69%, 73%, 82%, and 87% for the plain mortar, CCs-1, CCs-2, and CCs-3, respectively. These results suggest that the healing rate is improved by CCs. CCs-1, CCs-2, and CCs-3 improved the healing rate of the plain mortar by approximately ∆4%, ∆14%, and ∆18%, respectively. The healing rate of the CCs improved as the SC volume increased. The healing rate of CCs-3 was approximately twice that of CCs-1. What is noteworthy in these results is that the healing performance can be improved by CCs, and that the healing performance increases as the ratio of SCs among the CCs increases. These results were judged to be due to the mechanistic differences between MCs and SCs. As shown in Figure 1, SCs are hydrated by water; that is, SCs must be exposed to water for healing. However, as shown in Figure 2, MCs do not require water. Moreover, the core material of MCs is liquid. In this study, because we wanted to examine the healing performance of SCs when applied together with MCs, the specimens were cured in a water bath. Therefore, there may be a possibility of losing the core material of MCs in a water bath. In this study, this factor was not considered, and additional research is intended to be conducted in the future. Therefore, we believe that the improvement in the healing performance as the ratio of SCs increased was due to the above reasons. These results also show the same trend in Figure 8b. The healing rates corresponding to a 0.3 mm crack width were approximately 60%, 67%, 73%, and 79% for the plain mortar, CCs-1, CCs-2, and CCs-3, respectively. CCs-1, CCs-2, and CCs-3 improved the healing rate of the plain mortar by approximately ∆7%, ∆13%, and ∆19%, respectively. What is noteworthy in this result is that the plain mortar without CCs also showed a natural healing effect. Moreover, the greater the crack width, the smaller the healing range. In other words, this suggests that a certain proportion of cracks can be healed by the natural healing effect even in general structures, and that the natural healing performance can be improved through CCs. Our analysis showed that the natural healing effect was due to the unhydrated cement present in the cracks. These results indicate that the initial crack width can be reduced by the healing effect and prevented from expanding. In this study, the results were obtained for penetrating cracks. However, cracks in real environments gradually expand from microscopic cracks. Therefore, healing the initial microcracks means that it is possible to reduce large cracks as well, and the healing effect increases as the microcracks grow. In addition, CCs used to heal microcracks react only around microcracks, meaning that it is possible to respond to cracks that occur later. 

Figure 9a shows the decrease in the water flow rate of the specimen with cracks induced at 7 days of age, while Figure 9b shows the decrease in the water flow rate of specimens with cracks induced at 28 days of age. As shown by the evaluation in Figure 10, in the case of the plain mortar, there is a clear difference in the healing rate depending on the crack induction date. In general, the quality of cement mortar is based on the age of 28 days. Therefore, cracks were induced at the age of 28 days to observe the healing rate according to the healing period. However, for crack induction at the age of 28 days, the time required to consider up to 28 days of healing is 56 days, which is a significant amount of time. Accordingly, in this study, the correlation between the healing rates of the specimens in which the cracks were induced at 7 and 28 days of age was analyzed. These results can predict the healing rate for 28 days of age based on a test for only 7 days of age. The healing rates of the specimens with cracks induced at 7 days of age for the plain mortar, CCs-1, CCs-2, and CCs-3 were 69%, 73%, 82%, and 87%, respectively. The healing rates of the specimens with cracks induced at 28 days of age were 60%, 67%, 73%, and 79% for the plain mortar, CCs-1, CCs-2, and CCs-3, respectively. The difference in healing rates between crack induction at 7 and 28 days was 9%, 6%, 9%, and 8% for the plain mortar, CCs-1, CCs-2, and CCs-3, respectively. These results indicate that the healing rate of the specimen in which the crack was induced at the age of 28 days was decreased by approximately 8% pt compared to that of the specimen in which the crack was induced at the age of 7 days. Accordingly, several analyses should be performed. However, according to the results of this study, it is possible to roughly predict the healing rate at 28 days using the healing rate at 7 days after crack induction, given the error range.

#### 3.2.2. Crack Closure

Figure 10a,b show the monitoring results of the cracked surfaces. The cracks on the surface of the plain mortar were closed by the healing product after 28 days of healing; however, they were not completely repaired. Regardless of the composition ratio of CCs, all cracks were closed by the healing product. The differences in healing performance between CCs-1, CCs-2, and CCs-3 were negligible. In addition, no difference could be observed between the specimens where cracking was induced on the 7th and 28th days. The reason for this difference is due to the problem of internal cracks. Hence, no differences could be found in the surface cracks. In addition, healing products were equally observed, as shown in Figure 10c,d. Although internal cracks were not reviewed in this paper, an additional analysis of internal cracks is needed in the future. Additionally, the component analysis of the healing product is required.

## 4. Conclusions

In this study, as a basic study for the mixed use of solid capsules and microcapsules, the quality characteristics and healing performance of cement mortar using composite capsules were evaluated. The conclusions of this study are as follows:As a result of evaluating the characteristics of the fresh state of cement mortar, the flow decreased as the proportion of SCs among the CCs increased, and it was found that the flow decreased by up to 10%. However, in the case of flow loss, no significant effect could be observed.As a result of evaluating the characteristics of the hardening state of the cement mortar, CCs tended to reduce the compressive strength and flexural strength. In particular, as the proportion of SCs among the CCs increased, the strength values proportionally decreased, with a maximum decrease of 22%.As a result of evaluating the healing performance of the cement mortar, CCs tended to improve the natural healing performance. Furthermore, limited to the experimental conditions of this study, as the ratio of SCs among the CCs increased, the healing rate increased by up to 18%.Correlation analysis was performed between the healing rates at 7 and 28 days after crack induction. There was a difference of approximately 8% pt between the two. However, if expanded into a database through more experimental results, it is thought that the experimental period could be shortened through correlation.As a result of surface crack monitoring, it was found that CCs had a crack-closing effect by reducing the surface crack width, and the healing products were found to be acicular ettringite and hexagonal calcium hydroxide crystals. However, due to the limitation to surface cracks, no significant results could be obtained in the case of differences with respect to the crack induction time or the healing period.

## Figures and Tables

**Figure 1 materials-15-08819-f001:**
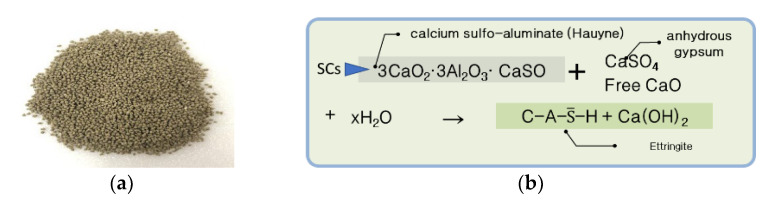
Solid capsules (SCs): (**a**) Sample of SCs. (**b**) Reaction mechanism of SCs.

**Figure 2 materials-15-08819-f002:**
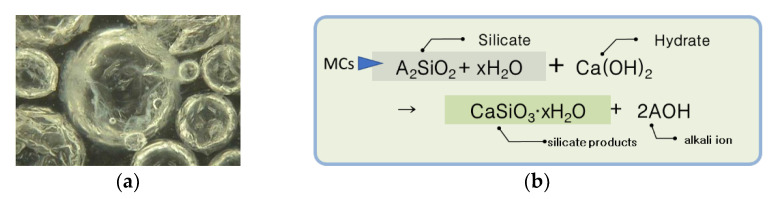
Microcapsules (MCs): (**a**) Sample of MCs (×300). (**b**) Reaction mechanism of MCs.

**Figure 3 materials-15-08819-f003:**
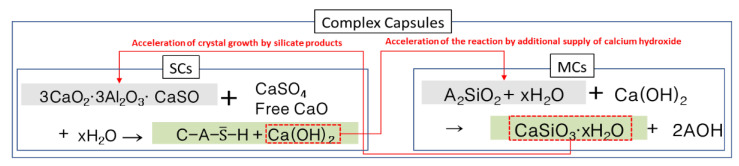
Reaction mechanism of the complex capsule.

**Figure 4 materials-15-08819-f004:**
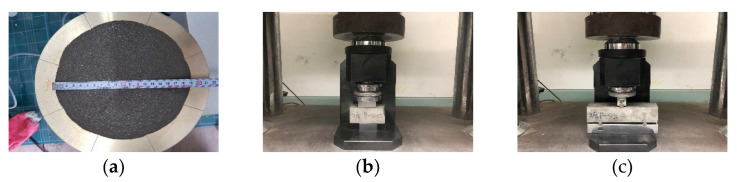
Reaction mechanism of the complex capsule: (**a**) flow test; (**b**) compressive strength test; (**c**) flexural strength test.

**Figure 5 materials-15-08819-f005:**
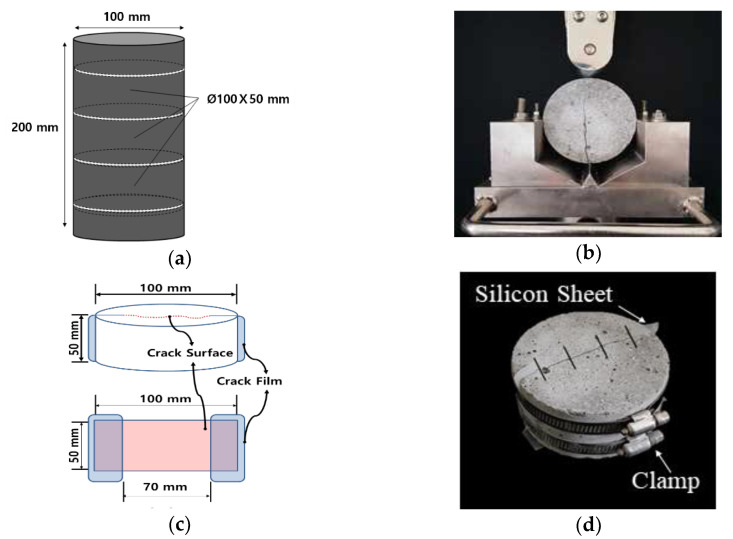
Preparation of cracked specimens: (**a**) slicing into three disc specimens, (**b**) split specimen, (**c**) crack induction, and (**d**) specimen bound together with two steel bands.

**Figure 6 materials-15-08819-f006:**
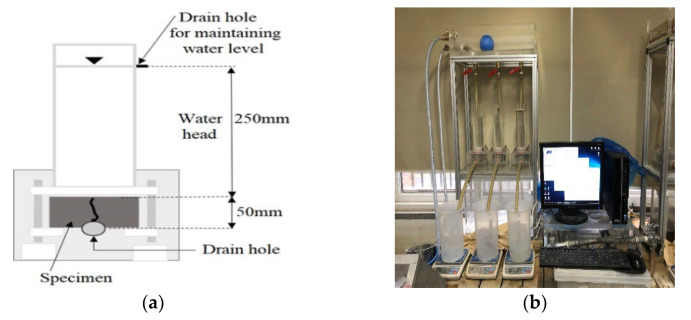
Water permeability test apparatus: (**a**) schematic diagram and (**b**) test setup.

**Figure 7 materials-15-08819-f007:**
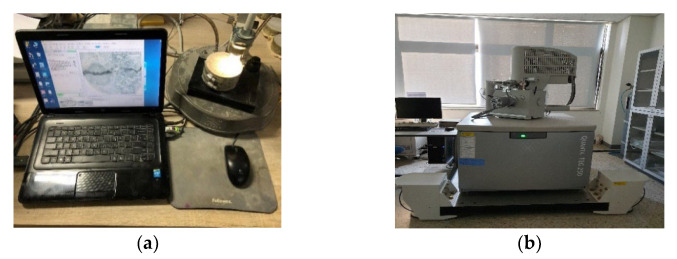
Healing product observation methods: (**a**) microscope and (**b**) SEM.

**Figure 8 materials-15-08819-f008:**
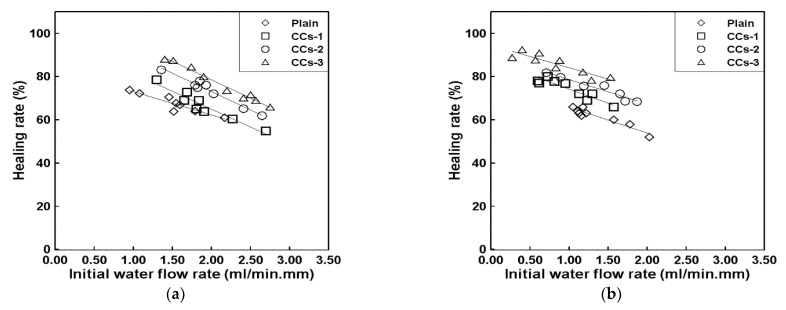
Correlation between the initial water flow rate and healing rate: (**a**) crack induction at 7 days; (**b**) crack induction at 28 days.

**Figure 9 materials-15-08819-f009:**
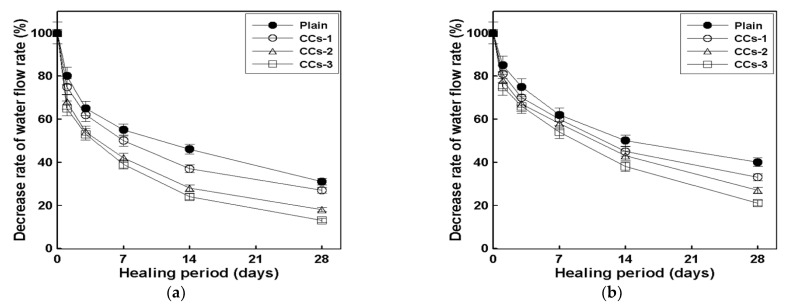
Correlation between the healing period and the decrease in the water flow rate: (**a**) crack induction at 7 days; (**b**) crack induction at 28 days.

**Figure 10 materials-15-08819-f010:**
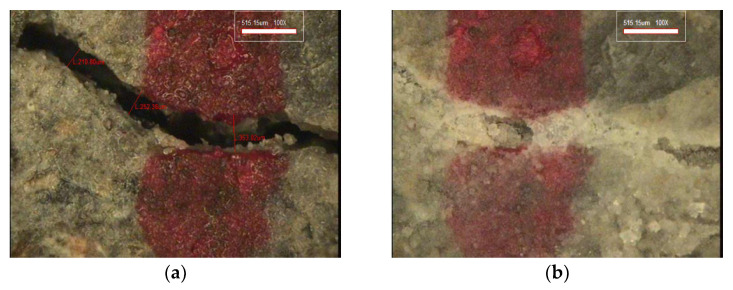
Crack monitoring results: (**a**) initial crack; (**b**) healed (healing period of 28 days); (**c**) SEM (×10,000); (**d**) SEM (×20,000).

**Table 1 materials-15-08819-t001:** Characteristics and manufacturing method of each capsule.

Type	SCs	MCs
Core materials	CaSO_4_:CSA = 7:3	K_2_SiO_3_:Na_2_SiO_3_:Li_2_SiO_3_ = 5:4:1
Wall materials	Polyurethane	Urea-formaldehyde
Special note	Coating of capsule wall after granulation of core material using coagulant	Silica coating (TEOS) to strengthen the capsule wall
Manufacturing method	Physical method [13]	Chemical method [14]
Size of capsules	600–850 μm	50–300 μm

**Table 2 materials-15-08819-t002:** Mixing ratio of the mortar.

Types	Water (W)	Binder (B)	Sand (S)	Complex Capsules (B×3%)		Ad.
				SCs	MCs	
Plain	0.4	1	2	0	0	0.6%
CCs-1	0.4	1	2	3	7	
CCs-2	0.4	1	2	5	5	
CCs-3	0.4	1	2	7	3	

**Table 3 materials-15-08819-t003:** Particle size distribution of ISO standard sand (ISO 679).

Size of Sieve (mm)	2	1.6	1.0	0.5	0.16	0.08
Cumulative residue of sieve (%)	0	7 ± 5	33 ± 5	67 ± 5	87 ± 5	99 ± 1

**Table 4 materials-15-08819-t004:** Fresh state characteristics of the cement mortars.

Types	Flow (mm)	Decrease in Flow Rate (%)	Flow Loss (%)		
			0 min	30 min	60 min
Plain	210	-	-	−11.9	−21.4
CCs-1	200	−4.8	-	−10.0	−20.5
CCs-2	195	−7.1	-	−10.3	−20.5
CCs-3	190	−9.5	-	−10.5	−21.1

**Table 5 materials-15-08819-t005:** Characteristics of the hardening state of the cement mortars.

Mix Type	Compressive Strength (MPa)				Flexural Strength (MPa)	
	3 Days	7 Days	28 Days	Decrease Rate (%)	28 Days	Decrease Rate (%)
Plain	36.3	45.5	50.0	-	10.4	-
CCs-1	35.3	43.1	48.0	−4.0	9.8	−5.1
CCs-2	32.7	40.0	45.0	−10.0	9.5	−8.1
CCs-3	29.7	35.0	39.0	−22.0	9.2	−11.6

## Data Availability

Not applicable.

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
