# Peer review of "An Experimental Study on the Healing Performance of Complex Capsules Using Multiphase Inorganic Materials for Crack Self-Healing of Cement Mortars"

_materials, 2022, doi:10.3390/ma15248819_

Round 1
Reviewer 1 Report
The topic and results shown are interesting, however, some aspects are not clear so I have the following questions.
1- Would be interesting to determine the setting time for pastes, and %porosity for mortars, both with CCs.
2- Page 3- lines 91 and 92: Please rectify the name for Fig. 1 and Fig. 2 (microcapsules).
3- Page 5- line 161- the phrase “Since the core material of the “is repeated on next line 162
4- Page 5- lines 163-164 – the phrase “was a liquid silicate material, the flow decreased due to a 163
slight quick-setting effect.” this is the same as line 162.
5- You can contrast the compressive strength results with physical properties (density and porosity) for mortars.
6- Page 6-line 181-182- please be more precise, CCs-1 (70%MCs-30%Cs). Or deleted the phrase, which is similar to 179-180. Please give a scientific reason for explaining this behavior, what happened in the microstructure of mortar?. Like flexural strength?
7- Page 7, line 219- pt? what is pt %?, please use International system notation. The same for page 8-line 243- 8 pt%, what is pt %?, please use International system notation.
8- What is the mechanism for the major success to healing rate in CCs-3 by CCs-1 ?. Why are major effect the MCs (silicates) than SCs (CSA-CASO4)
9- Fig 11b, what is ffd?, Does this image corresponds to Plain?.
10- Page 9-line 275 to 282- conclusion 1- many words are repeated, deleted and rectify
Author Response
November 19, 2022
Dear Reviewer,
Thank you very much for your valuable comments. I, with my co-authors, carefully revised the manuscript entitled “An Experimental Study on the Healing Performance of Complex Capsules using Multiphase Inorganic materials for Crack Self-healing of Cement Mortars” according to the reviewer’s comments as follows.
Comment 1: Would be interesting to determine the setting time for pastes, and %porosity for mortars, both with CCs.
Response 1: Thank you for your review. Our authors did not understand what your first comment meant. Although this comment was not reflected in the manuscript, we will analyze it through a new approach in the future.
Comment 2: Page 3- lines 91 and 92: Please rectify the name for Fig. 1 and Fig. 2 (microcapsules).
Response 2: It was confirmed that many of the caption numbers in tables and figures throughout the manuscript were incorrect, and supplemented throughout the manuscript.(Fig.1 and Fig.2, Changed lines 112, 113)
Comment 3: Page 5- line 161- the phrase “Since the core material of the “is repeated on next line 162
Response 3: Sentence errors were identified. The paragraph was rewritten including the error part.( Changed page 7, lines 197-204)
Comment 4: Page 5- lines 163-164 – the phrase “was a liquid silicate material, the flow decreased due to a 163, slight quick-setting effect.” this is the same as line 162.
Response 4: Same as response 3 to comment 3.
Comment 5: You can contrast the compressive strength results with physical properties (density and porosity) for mortars.
Response 5: Same as response 1 to comment 1. However, we will try various analyzes in the future. thanks for your comment. This section has been newly written.(Changed page 7, lines 207-221)
Comment 6: Page 6-line 181-182- please be more precise, CCs-1 (70%MCs-30%Cs). Or deleted the phrase, which is similar to 179-180. Please give a scientific reason for explaining this behavior, what happened in the microstructure of mortar?. Like flexural strength?
Response 6: This section has been newly written.(Changed page 7, lines 207-221)
Comment 7: Page 7, line 219- pt? what is pt %?, please use International system notation. The same for page 8-line 243- 8 pt%, what is pt %?, please use International system notation.
Response 7: 'pt.%' is an abbreviation for 'point %'. However, 'pt' was deleted from the manuscript because it is not suitable for international notation. (Changed page 8, lines 240, 256)
Comment 8: What is the mechanism for the major success to healing rate in CCs-3 by CCs-1 ?. Why are major effect the MCs (silicates) than SCs (CSA-CASO4)
Response 8: The content was mentioned in the manuscript and supplemented.(Fig.1, Fig.2, Table. 1, Line 242-255, “Abstract”, “Introduction” )
Comment 9: Fig 11b, what is ffd?, Does this image corresponds to Plain?.
Response 9: The caption of the figure has been corrected.( Changed Figure 10 (b), Line 309)
Comment 10: Page 9-line 275 to 282- conclusion 1- many words are repeated, deleted and rectify.
Response 10: The "Conclusion" section has been revised and supplemented overall.(Changed pages 10-11, Line 311-342.)
Thank you for reviewing the paper. We will finalize the manuscript by reflecting the opinions of all reviewers along with your comments.

Reviewer 2 Report
1. Line 8 - The beginning of the abstract is repeated: "In this study".
2. Insert the search issue/justification into the summary. The summary is already starting right with the goal.
3. Line 8 - Explain what Is Ad (Bx%) in Table 1.
4. Line 95 - What is sand granulometry?
5. Line 105 - Briefly explain the method and equipment used to evaluate the fluidity of mortars.
6. Line 109 – Describe how many specimens have been tested and the model and brand of equipment used for compression strength tests.
7. Line 151 - The source of Fig. 4th is too small and cannot read.
8. Line 161 and 162 – The words "Since the core material" are repeated in the paragraph.
9. Review the numbering of the figures in the text. In many of them, calls in the text do not represent the correct numbering of the figure.
10. Line 182 – The phrase "The decrease rate of compressive strength increases as the volume of SCs 182 among the CCs increases. " You're acting strange. In fact, it was found that the increase the amount of SCs and CCs decreased the compressive strength of mortars.
11. Line 189 – There is a strange symbol before CCs.
12. Line 200 - Insert unit from 1.0 to 1.8.
13. Line 276 – Loss of Since the core material of the Since the core material of the MCs. Repeated words in the text.
14 Figure 9 shows the taxa of self-healing. In the methodology of the work, it is not clearly described how this measure was determined.
15. Line 256 – The description of the ages evaluated in the trials should go to the methodology of the work.
Reviewer 3 Report
The manuscript is not prepared carefully. There are many rookie mistakes. Hence, I must reject the manuscript for the publication.
1. Many important figures and equations are missing. For example: Fig. 1, Eq. 1, etc.
2. There are many grammatical mistakes. For example: Line 8, Line 277-281, etc.
3. There are many mistakes for the figure number. For example: Line88, Line 89, Line 149, Line 157, Line 165, etc.
4. The “Introduction” section should be rewritten. The literature review is too oversimplified. The objectives and innovations of this study are also not clear.
5. The “Conclusion” section should be refined.
6. Why the SEM results at 3-day and 7-day are missing. At least, the SEM results at 7-day should be supplemented.
Round 2
Reviewer 1 Report
The corrections sent in the first round were attended by the authors, and the document is now clearer and more organized. Although the authors did not send a Clean Version, to verify that there were no organizational errors.
It can be accepted for publication.
Reviewer 3 Report
All the comments have been addressed.